# Rethinking of Environmental Health Risks: A Systematic Approach of Physical—Social Health Vulnerability Assessment on Heavy-Metal Exposure through Soil and Vegetables

**DOI:** 10.3390/ijerph182413379

**Published:** 2021-12-19

**Authors:** Jun Yang, Silu Ma, Yongwei Song, Fei Li, Jingcheng Zhou

**Affiliations:** 1Research Center for Environment and Health, Zhongnan University of Economic and Law, Wuhan 430073, China; jun.yang@zuel.edu.cn (J.Y.); songyongwei@zuel.edu.cn (Y.S.); lifei@zuel.edu.cn (F.L.); 2Wuhan Planning and Design Company, Wuhan 430014, China; wpdc@wpdc2012.com

**Keywords:** heavy-metal exposure, population environmental health risk, physical vulnerability, social vulnerability, index evaluation method, village scale

## Abstract

In the field of environmental health risk assessment and management research, heavy metals in soil are a constant focus, largely because of mining and metallurgical activities, and other manufacturing or producing. However, systematic vulnerability, and combined research of social and physical vulnerability of the crowd, have received less attention in the research literature of environmental health risk assessment. For this reason, tentative design modelling for comprehensive environmental health vulnerability, which includes the index of physical and social vulnerability, was conducted here. On the basis of experimental data of heavy-metal pollution in soil and vegetables, and population and societal survey data in Daye, China, the physical, social, and comprehensive environmental health vulnerabilities of the area were analyzed, with each village as an evaluation unit. First, the polluted and reference areas were selected. Random sampling sites were distributed in the farmland of the villages in these two areas, with two sampling sites per village. Then, 204 vegetable samples were directly collected from the farmland from which the soil samples had been collected, composed of seven kinds of vegetables: cowpea, water spinach, amaranth, sweet potato leaves, tomato, eggplant, and pepper. Moreover, 400 questionnaires were given to the local residents in these corresponding villages, and 389 valid responses were obtained. The results indicated that (1) the average physical vulnerability values of the population in the polluted and reference areas were 3.99 and 1.00, respectively; (2) the village of Weiwang (WW) had the highest physical vulnerability of 8.55; (3) vegetable intake is exposure that should be paid more attention, as it contributes more than 90% to physical vulnerability among the exposure pathways; (4) arsenic and cadmium should be the priority pollutants, with average physical vulnerability value contributions of 63.9% and 17.0%, respectively; (5) according to the social vulnerability assessment, the village of Luoqiao (LQ) had the highest social vulnerability (0.77); (6) for comprehensive environmental health vulnerability, five villages near mining activities and two villages far from mine-affected area had high physical and social vulnerability, and are the urgent areas for environmental risk management. In order to promote environmental risk management, it is necessary to prioritize identifying vulnerable populations in the village-scale dimension as an innovative discovery.

## 1. Introduction

Heavy metals, as a representative type of environmental pollution [1] via different environmental media, such as air, water, and soil, affect and harm individuals. Environmental chemical pollutants and human exposure patterns are often the focus of environmental health research [2]. By the validation of Public Health Exposome, an evidence based frame work can be used to research the relationships between differential levels of exposure at critical stages, personal health outcomes, and health disparities at a population level (e.g., community level) [3,4]. In addition to environmental pollution exposure, research showed that social factors such as poverty [5], lack of education, and poor living conditions [6] may be connected with the existence of environmental health disparities [7,8]. For this reason, the U.S. Environmental Protection Agency (USEPA) proposed the Framework for Cumulative Risk Assessment to evaluate cumulative health risks from both chemical mixtures, and a combination of chemical and nonchemical stressors to communities [9,10,11]. Researchers in the field of environmental justice (EJ) draw from the concept of environmental health vulnerability [12] and social vulnerability [13,14]. They also advocate that both environmental chemical pollutants and social factors should be considered in regulatory procedures and decision-making activities.

Overall, the main approaches used to evaluate the cumulative health impact of pollution exposure and social stressors have different evaluation systems and frameworks because of their different emphases. Some quantitative or semi-quantitative methods, such as health risk assessment [15], health impact assessment [16], and burden of disease [17,18], mainly focus on pollutant exposure, but weaken the impact of social factors. In order to evaluate environmental health vulnerability, a series of evaluation index systems, namely, Cumulative Environmental Hazard Inequality Index (CEHII) [19,20], the Environmental Justice Screening Method (EJSM) [21,22] with its associated Proposed Climate Change Vulnerability Screening Method (CCVSM) [23], the Cumulative Environmental Vulnerability Assessment (CEVA) [24], the California Community Environmental Health Screening Tool (CalEnviroScreen) [25], and Public Participatory Geographical Information Systems (PPGIS) [26], were put into practice with the help of GIS tools. Due to the sophisticated perspectives of social determinants, environmental exposures and health disparities of population or communities, the algorithm-based Scalable Combinatorial Tools also supported to reveal the causal mechanisms and environmental contexts beneath the health disparities [27]. Moreover, by using the novel exposome and a graph-theoretical toolchain, exposures with disparities of gender and race even considering the mortal health diseases or health risks of humans in different territories can be analyzed [28]. The indicators of such methods are often used at the state or provincial levels; thus, evaluation units are medium-level-oriented, which hint at a much finer level being preferable, such as villages or particularly tantamount areas, to accurately reflect small-scale differences.

Pollution affects human health, particularly reproductive health [29,30]. Heavy metals are considered to be critical pollutants, i.e., cadmium and arsenic, of exposure sources, resorption pathways, and organ damage, have attracted scientific attention for decades [31,32,33,34]. In the previous relevant study of lead for example, a quantitative samples addressing the pathways, bare soil to Pb hazards and even site investigation of individuals home environment, should be identified carefully [35]. In China, soil contamination by toxic metals is prevalent and serious due to the rapid urban development and substantial productive industrial activities [36,37,38]. Toxic metal pollution is a major health threat to human beings because of its persistent toxicity and not being easily biodegradable [39,40,41]. Exposure to several heavy metals, i.e., lead, arsenic, and cadmium, is a risk factor for cancer [42,43,44,45] and the development of several other diseases, especially cardiovascular, kidney [46], nervous system, blood, and bone [47]. Daye is a city that has been an important metal ore concentration area in Central China with a long mining history of 3000 years. For this reason, cumulative heavy-metal contamination caused by persistent mining activities has attracted extensive attention. For China’s rural areas, the implementation of policies often depends on the smallest administrative unit, so small-scale research is more conducive to the identification of vulnerable groups in the region, and the formulation and implementation of detailed environmental risk management policies.

The purpose of this research was to identify priority protection objectives with high health vulnerability among 16 villages in Daye through environmental health vulnerability assessment, and provide important information for environmental risk management. First, we investigated the pollution situation in the study areas, and analyzed the pollution exposure characteristics and pollution hazard level of the local residents. Then, we collected exposure parameters and socioeconomic data, and analyzed the social vulnerability. Lastly, we evaluated environmental health vulnerability, which is social vulnerability assessment combined with the hazard index of heavy-metal pollution.

## 2. Materials and Methods

### 2.1. Study Area

In central China, the city of Daye is located in the southeast of Hubei province. Sixteen villages in Daye were selected as the study areas, nine of which are mine-affected (polluted) areas in the northeast of Daye, while seven are relatively far from the mining (reference) area in the west of Daye. There are 32 random sampling sites distributed in the farmland of the selected villages, with two sampling sites per village. In the polluted areas, there were 18 sampling sites in 9 villages: Weiwang (WW, sampling sites 1 and 2), Jinqiao (JQ, sampling sites 3 and 4), Wangjiazhuang (WJZ, sampling sites 5 and 6), Luoqiao (LQ, sampling sites 7 and 8), Huajing (HJ, sampling sites 9 and 10), Chunguang (CG, sampling sites 11 and 12), Changle (CL, sampling sites 13 and 14), Guantang (GT, sampling sites 15 and 16), and Tuannao (TN, sampling sites 17 and 18). In the reference area, there were 14 sampling sites in 7 villages: Shangwang (SW, sampling sites 19 and 20), Zhushan (ZS, sampling sites 21 and 22), Mingshan (MS, sampling sites 23 and 24), Fandao (FD, sampling sites 25 and 26), Wuduan (WD, sampling sites 27 and 28), Yangqiao (YQ, sampling sites 29 and 30), and Shangzhuang (SZ, sampling sites 31 and 32). Villages in these two study areas were selected considering the inhabitants’ settlement and distribution. All geographical locations of the sampling points in the polluted and reference areas are shown in Figure 1.

### 2.2. Data Sources

#### 2.2.1. Heavy-Metal Pollution Data

Content analysis of heavy metals in soil and common vegetables in the study area was conducted to assess physical vulnerability. According to the Chinese Technical Specification for Soil Environmental Monitoring (HJ/166-2004), 32 surface soil samples (1 kg each and from the top 0 to 20 cm layer at the sampling sites) were collected. In total, 204 vegetable samples were directly collected from the farmland from which the soil samples were collected comprising 7 kinds of vegetables: cowpea (*Vigna unguiculata* (*Linn.*) *Walp*), water spinach (*Ipomoea aquatica Forsk*), amaranth (*Amaranthus tricolor* L.), sweet potato leaves (*Ipomoea batatas Lam*), tomato (*Lycopersicon esculentum Miller*), eggplant (*Solanum melongena Linn*), and pepper (*Capsicum annuum Linn.* var. *gros-sum* (L.) *Sendt*). A large number of local residents buy rice and meat from markets, and consume vegetables that they grow in their own field. Moreover, rice mainly refers to paddy rice, with supplements of wheat flour, and meat mainly includes pork, chicken (as food), mutton, and beef. The sample pretreatment and analysis were based on the method designed in our previous research, which was analyzed in detail by Jun et al. [48].

#### 2.2.2. Population Survey Data

According to the population situation in the Daye area, we designed a questionnaire of its population parameters. The questionnaire mainly included basic information, exposure parameters, personal habits, and health information. Questions on basic information of the respondents regarded age, height, weight, occupation, education level, and income. Dietary exposure parameters included the types of vegetables grown at home, the types of vegetables consumed daily, the consumption of fresh vegetables, and other dietary behavior parameters. Habit and health information included questions about smoking, illness, sleeping time, working hours, and work intensity.

We distributed 400 questionnaires to the local residents in these 2 areas, and 389 valid responses were obtained. After data input, PASW Statistics, version 25.0 (SPSS Inc., Chicago, IL, USA) was used for statistical analysis.

### 2.3. Vulnerability Evaluation Method

Vulnerability refers to the threat to which an area is exposed from the properties of involved chemical agents, the ecological situation of the community, and its general state-of-emergency preparedness at a given point in time [49]. For environmental risk management, the assessment of environmental health vulnerability considers two aspects: first, heavy-metal exposure and the possible health of communities or groups, namely, physical vulnerability [50]; and second, the comprehensive measurement of the sensitivity, and the coping ability, adaptability, and resilience of communities in the face of threats, namely, social vulnerability [51].

Integrated population environmental health vulnerability is divided into two aspects, physical and social vulnerability. We examined how to evaluate the above-mentioned vulnerability. Hence, an analytical framework considering population environmental health vulnerability assessment on heavy-metal exposure was designed, as shown in Figure 2. Oral intake and soil exposure are two specific pathways to conduct this research. Moreover, a preliminary investigation found that the majority of local residents eat vegetables growing in local soil, which may have been affected by heavy-metal exposure, and they mainly grow them for self-consumption. Therefore, what is more important is that contaminated soil and vegetable intake are correlated.

#### 2.3.1. Assessment of Physical Vulnerability

The sensitivity of the human body to different heavy-metal species is different. Exposure to various heavy metals cannot be simply added up, so it is not appropriate to measure physical vulnerability by simply using the heavy-metal concentration in environmental media or average daily doses.

The index system of pollution hazard in health risk modelling provided by USEPA is used as a measure of physical vulnerability. The hazard index (HI) for the health risk of a variety of heavy metals is calculated by Equation (1), and the corresponding dose received through each of the four pathways was evaluated by Equations (2)–(5).
(1)$$\begin{array}{ll} \mathrm{HI} & ={\mathrm{HQ}}_{v}{+\mathrm{HQ}}_{o}{+\mathrm{HQ}}_{d}{+\mathrm{HQ}}_{i}\\ & =\frac{{\mathrm{ADD}}_{v}}{{\mathrm{RfD}}_{v}}+\frac{{\mathrm{ADD}}_{o}}{{\mathrm{RfD}}_{o}}+\frac{{\mathrm{ADD}}_{d}}{{\mathrm{RfD}}_{d}}+\frac{{\mathrm{ADD}}_{i}}{{\mathrm{RfD}}_{i}}\; \end{array}$$
(2)$${\mathrm{ADD}}_{v}=\frac{C_{v}\times {\mathrm{IR}}_{v}\times {\mathrm{EF}}_{v}\times \mathrm{ED}}{\mathrm{BW}\times \mathrm{AT}}$$
(3)$${\mathrm{ADD}}_{o}=\frac{C_{s}\times {\mathrm{IR}}_{o}\times \mathrm{CF}\times \mathrm{EF}\times \mathrm{ED}}{\mathrm{BW}\times \mathrm{AT}}$$
(4)$${\mathrm{ADD}}_{d}=\frac{C_{s}\times \mathrm{ABS}\times \mathrm{SA}\times \mathrm{AF}\times \mathrm{EF}\times \mathrm{ED}}{\mathrm{BW}\times \mathrm{AT}}$$
(5)$${\mathrm{ADD}}_{i}=\frac{C_{s}\times {\mathrm{IR}}_{b}\times \mathrm{EF}\times \mathrm{ED}}{\mathrm{PEF}\times \mathrm{BW}\times \mathrm{AT}}$$
where HQ_v_, HQ_o_, HQ_d_, and HQ_i_ are the hazard quotients caused by the four pathways of vegetable intake, soil ingestion, dermal contact, and inhalation, respectively; RfD_v_, RfD_o_, RfD_d_ and RfD_i_ are the corresponding reference doses for each heavy metal through one of the four pathways, respectively, as shown in Table 1; ADD_v_, ADD_o_, ADD_d_, and ADD_i_ are the average daily doses from vegetable intake, soil ingestion, dermal contact, and inhalation, respectively (mg/kg·day); C_v_ is measured by the average heavy-metal content of vegetables sampled in each village (mg/kg); C_s_ is measured by the average heavy-metal concentration of soils from two sample sites in each village (mg/kg); IR_v_, refers to the intake rate of vegetable (mg/day); IR_o_ refers to ingestion (mg/day); IR_b_ refers to inhalation rate of soil (m^3^/day); EF is exposure frequency (day/year); ED is exposure duration (year); BW is the average body weight of the exposed individual (kg); AT is the averaged contact time (day); PEF is the particle emission factor (m^3^/kg); SA is the exposed skin surface area (cm^2^); AF is adherence factor (mg/m^2^·day); and ABS is the dermal absorption factor (unitless). Detailed information of the above-mentioned and probabilistic parameters can be found in Jun et al. [48], and Fei et al. [52]. Parameters and their values, used to evaluate physical vulnerability in the above equations, are summarized in Table 1.

#### 2.3.2. Social Vulnerability Index System

For a human community, health outcomes are impacted by the relationships among measures of socioeconomic level (ability to respond or recover), receptor characteristics (measures of potential vulnerability), and population self-sensitivity. On the basis of a series of social vulnerability studies and the local conditions of the study area, 10 indicators were selected and divided into 3 categories. The important relationship among the indices was established by expert scoring, and the weights of these 10 indices were determined with the support of analytic hierarchy process theory. The experts who participated in the research had a wide range of health risks and a basic understanding of receptor vulnerability. Ultimately, social vulnerability scores were calculated and used to assess environmental health vulnerability.

The explanation and quantification of indicators are shown in Table 2. It is more appropriate for a region to quantify indicators by using the proportion of people with higher vulnerability. People with higher incomes feel more satisfied [53] and can afford more local daily consumption. Therefore, we measured the economic level of the research areas by the proportion of local people who have reached the per capita disposable income in Daye. Sleeping time and working time reflect people’s social pressure [54], and reasonable sleeping and working time (8 h) are conducive to health [55]. The psychological and physiological conditions of a person after 13 years tend to be mature [56], while the Chinese legal retirement age is 65 years old in common [57]. People aged 14–65 are in good condition and usually able to resist potential social or physical threats. People under 14 or over 65 years old in the research areas, on the other hand, are too young or too old to face off threats. Therefore, under 14 or over 65 years are regarded to be ages with much higher vulnerability. It is easy to identify the vulnerable population in various indicators according to previous research, such as educational level, occupation [58], working environment, labor intensity, gender [59], and disease.

#### 2.3.3. Environmental Health Vulnerability Assessment

Multiple models are widely used in some regional vulnerability models and cumulative impact frameworks. However, for a group of population living in particular area rather than individual research subjects, it is interesting to simultaneously consider environmental health risks with physical and social vulnerability. The interplay of group behavior, habits, and population characteristics should not be ignored even in research of environmental health risk, which typically reveals social vulnerability. The environmental health vulnerability index is obtained by multiplying the quantitative physical-vulnerability value with the social vulnerability index, which can synthetically reflect the vulnerability of villages.

Vulnerability thus consists of physical and social vulnerability. On the basis of these two dimensions, a four-quadrant chart is presented in Figure 3 for confirming the grade of overall vulnerability. Physical vulnerability is defined as a threshold of when the pollution hazard index is less or more than 1.0. According to relevant USEPA research, the pollution hazard index is lower than 1.0. In addition, for social vulnerability, appropriate discontinuous scores of top to bottom limitation are 0 to 1.0. However, integrating discrete values with continuous values may cause conflict in general vulnerability assessments, so social vulnerability is defined as the threshold when the score of social vulnerability is less or more than 0.5 to 1.0. Therefore, a quantified conceptual frame was applied to classify vulnerability into four categories, and different villages were sorted into different areas of the quadrant.

## 3. Results

### 3.1. Physical Vulnerability Assessment

According to the physical vulnerability evaluation method, the physical vulnerability value was calculated using Equations (1)–(5), and the evaluation results of 16 villages in the study area are shown in Table 3. The pollution hazard index in the table is the quantitative value of physical vulnerability, which is presented according to different exposure routes and different kinds of heavy metals.

The physical vulnerability of different villages was in the order of WW > JQ > GT > WJZ > LQ > HJ > CL > CG > ZS > YQ > TN > SZ > SW > WD > FD > MS. The physical vulnerability of the villages in the polluted area was significantly higher (*p* < 0.05) than that in the reference area through each exposure route. The calculated average values of population physical vulnerability in the polluted and reference areas were 3.99 and 1.00, respectively. Values of physical vulnerability of all villages in the polluted area and ZS, YQ, and SZ in the reference area were significantly higher than the upper threshold of 1.0, among which the highest was 8.55. From the perspective of physical vulnerability, ZS, YQ, and SZ in the reference area were also key villages of heavy-metal contamination.

The pollution hazard index of each exposure route showed the order of vegetable intake > soil ingestion > dermal contact > inhalation. More than 90% of the pollution hazard index was contributed by vegetable intake for every village in the study areas.

The average physical vulnerability value contribution of different heavy metals showed the order of A_s_ (63.9%) > C_d_ (17.0%) > C_u_ (8.7%) > Z_n_ (6.8%) > P_b_ (3.2%). In fact, heavy metals with a major contribution to the physical vulnerability were not the same in different exposure routes. As and Cd similarly contribute to physical vulnerability by the pathway of vegetable intake, reaching 36.5% and 37.3%, respectively. Additionally, A_s_ contributed the most among other metals to the physical vulnerability value through the exposure routes of soil ingestion and inhalation, reaching 76.9% and 87.8%, respectively. There was little difference in the contribution of P_b_, A_s_, and C_d_ through dermal contact, reaching 45.9%, 29.5%, and 23.0%, respectively.

### 3.2. Social Vulnerability Assessment

The higher the score of social vulnerability is, the weaker the ability of the population to resist and protect themselves against potential pollution hazards and other pressure disturbances. That is to say, the group with higher social vulnerability may lose more when facing the same pollution threat. Scores of each index in the social vulnerability evaluation system and the composite score are shown in Table 4.

The total social vulnerability scores of 16 villages from high to low are as follows: LQ (0.770), FD (0.690), SW (0.688), MS (0.639), ZS (0.592), GT (0.543), TN (0.499), CL (0.490), JQ (0.464), WJZ (0.459), YQ (0.453), WW (0.360), WD (0.348), SZ (0.342), HJ (0.334), and CG (0.188). From the total score of each village, LQ had the highest social vulnerability, and CG had the lowest social vulnerability. Indicators of self-sensitivity show the local residents’ own sensitivity to pollution hazards. The higher the SS composite score is, the higher the sensitivity, and the more likely it is to have negative health effects in the face of pollution threats.

The score of social and economic conditions reflects the occupancy of social resources. The higher the social vulnerability score is, the fewer social resources the village requires. The population of MS needed the fewest social resources among 16 villages, with the highest SEC composite score of 0.31. The indicators of receptor behavioral characteristics represent the social life stress of the local population. The higher the score is, the greater the pressure they are under. The population of SW was burdened by the heaviest social stress among the 16 villages with the highest BE composite score of 0.28.

In fact, the correlation analysis of each index score in the social vulnerability evaluation system could roughly reflect the local population and social characteristics. SPSS was used to analyze the correlation of each index, and results were shown in Table 5. The population of LQ had the highest sensitivity among the 16 villages, with the highest SS composite score of 0.29.

The correlation of some indicators related to our research was consistent with the results of large sample data analysis in many studies. As a result, a significant positive correlation was found between the proportion of people with a low educational background and the proportion of people with income lower than the local average income (*p* = 0.034 < 0.05). This result is consistent with the research of Tachibanaki [60] and Li [61]. There was significant positive correlation between the proportion of vulnerable age groups and the proportion of people suffering from diseases (*p* = 0.048), which is consistent with the finding that age and gender are both risk factors for chronic diseases [62]. There is significant positive correlation between the proportion of people with a bad working environment and the proportion of people with excessive labor intensity (*p* = 0.007 < 0.05). There was significant positive correlation between the proportion of women and the proportion of vulnerable age groups (*p* = 0.020 < 0.05). These two phenomena are related to the survey area. Because the survey area is rural, and its geographical location is remote, local adult males may go out to work more, so the proportion of females and the number of vulnerable age groups in the 16 surveyed villages was higher. In Chinese rural areas, there are many people engaged in heavy manual labor in harsh working environments.

### 3.3. Comprehensive Environmental Health Vulnerability Assessment

The comprehensive score of environmental health vulnerability was obtained by multiplying the physical and social vulnerability scores, and villages were marked in the quadrant (Figure 4) according to the hierarchical evaluation of physical and social vulnerability.

The environmental health vulnerability of the 16 villages ranged from high to low was as follows: JQ (0.40), WW (0.36), LQ (0.35), GT (0.26), WJZ (0.21), CL (0.14), ZS (0.12), HJ (0.10), YQ (0.08), SW (0.07), TN (0.06), CG (0.05), FD (0.05), SZ (0.04), MS (0.04), and WD (0.03).

Figure 4 shows that the evaluated villages were classified into four categories by the hierarchical evaluation matrix: WD had low physical vulnerability and low social vulnerability; MS, SW, and FD had low physical vulnerability and high social vulnerability; SZ, CG, HJ, and WW had high physical vulnerability and low social vulnerability; YQ, TN, ZS, and FD had high physical vulnerability and low social vulnerability. Five villages near mining activities (JQ, LQ, GT, WJZ, CL, and TN) and two villages (ZS and YQ) far from the mining-affected area have high physical vulnerability and high social vulnerability, which are the urgent areas in environmental health risk management.

## 4. Discussion

### 4.1. Villages with High Values of Physical Vulnerability and Social Vulnerability Are Screened

Soil and vegetable pollution data were obtained through experimental detection, and population and social characteristic data were obtained through investigation, with the environmental health vulnerability evaluation performed with the tentative comprehensive evaluation method. Results indicated that the average physical vulnerability values of the population in the polluted and reference areas were 3.99 and 1.00, respectively. For villages with sampling sites, WW had the highest physical vulnerability value of 8.55. Considering exposure pathways, vegetable intake is the exposure type that should be paid more attention, as it contributes more than 90% to physical vulnerability among the four exposure pathways. Arsenic and cadmium should be the priority pollutants, with average physical vulnerability value contributions of 63.9% and 17.0%, respectively. According to the social vulnerability assessment, LQ had the highest social vulnerability with a value of 0.77. Comprehensive environmental health vulnerability assessment showed that five villages near mining activities (JQ, LQ, GT, WJZ, CL, and TN) and two villages (ZS and YQ) far from the mine-affected area are urgent areas in environmental risk management, with high physical and social vulnerability.

### 4.2. Advantage of Establishing Physical and Social Vulnerability

For environmental health risk, there is an entire solving structure presented by EPA or CDC authorities in the U.S., which is widely recognized in the era of scientific research to receive hazard index of physical pollutants. We omitted the process and focused more on introducing social rather than physical aspects. Social and physical vulnerability contribute to the population’s environmental health risk. On the basis of the results, social and physical vulnerability in research of heavy-metal exposure leading to environmental health risk are not significantly related with each other. Considering divergent factors of population vulnerability in a certain area, population distribution, regional development, and mining activities are likely to influence these two types of vulnerability. Our previous study showed that heavy-metal exposure in soil is a cause of environmental health risk, particularly in a long-term mining city such as Daye, China.

From our research, we learned that risk and vulnerability likely band with each other, even in the aspect of environmental health risks. The result shows that social and physical vulnerability cause significant population environmental health risk. It is appropriate not to add social to physical vulnerability to solve the calculation of the environmental health risk modelling, but to combine the two types of vulnerability and calculate each type individually as presented above. Furthermore, population environmental health risks, which focus the research subject on a group of people or residents in a certain place, need to be carefully considered with the social conditions and characteristics of the group.

It is innovative to create a systematic approach with social and physical vulnerability to deal with population environmental health risk that uses the hazard index for the evaluation of physical vulnerability and creates a novel social vulnerability index system to evaluate the later vulnerability on the basis of a large-scale of door-to-door survey of residents of 24 villages in the research area. Therefore, the research would not assume individuals to be one subject, but consider a number of local residents to be a population. The critical problem lies in the fact that it is inadvisable to ignore population distribution and characteristics in the research of population vulnerability and the environmental health of local residents.

### 4.3. Selected Village-Scale Dimension Applied to Draw Specific Contrasting Differences

Results indicated that long-term mining activities may influence a percentage of the population and in a large range of areas. According to the researching dimension, village-scale areas are specific and with significant differences, which helped us to confirm the suitability of geographic division between the polluted the reference areas.

By filtering the results of physical vulnerability assessment in the case study, villages in the polluted area are more likely to be exposed to environmental health risk than the ones in the reference area. Sample analysis and health risk calculation revealed that the adjacent contiguous villages of WW, WJZ, JQ, and LQ, had a high value of HI (HI > 3.50). The adjacent contiguous villages of MS, FD, and WD, on the other hand, had a lower value of HI (HI < 1.00). An adjacent–contiguous feature indicates a spatial characteristic of physical vulnerability.

Regarding social vulnerability, the adjacent contiguous villages of SW, ZS, MS, and FD in the reference area had a higher assessment value (score > 0.59). The adjacent contiguous villages of LQ, TN, and JQ in the reference area had a higher assessment value (score > 0.45). However, the difference between physical and social vulnerability is that the former one is specific and the latter is universal, with or without the environmental health risk problem. The population’s social vulnerability widely exists according to the result (score > or =0.19), but comprehensive population vulnerability is more meaningful with both the social and the physical aspect. Furthermore, when considering population social vulnerability, there is an adjacent–contiguous feature that indicates a spatial characteristic of social vulnerability without zoning differences, and this feature was notable in both the reference and the polluted area.

### 4.4. Research Novelty and Limitations

There is a continuous and cumulative risk of heavy-metal exposure in mining areas that can be obtained both from sample analysis and questionnaire survey. Social vulnerability indicates people’s livelihood, population effects, group behaviors, and population differences under a certain stress or risk or multiple sources, which should not be neglected in environmental health risk assessment activities. The cohorts, as a large number of residents living locally in the two studied areas, considering exposure to environmental health risk of heavy-metal exposure, had significantly different social vulnerability. A nonrandomized control study of the polluted and reference areas by the village-scale zoning is applicable.

Different environmental protection measures and risk management control strategies should be adopted more meticulously for different vulnerability characteristics. Social vulnerability is related to many more factors, and some cannot be entirely obtained, such as the local educational background structure, income level, the proportion of people in industries with large opportunities of direct contact with pollutants such as mining, construction, and agricultural planting.

The research area in the case study belongs to rural areas of the city of Daye, where the economic development level is lagging behind that of cities and towns. In addition, a major feature of the local rural areas is that there have been more mining activities resulting in a higher proportion of the local population with industrial employment and mining as the main occupation. The reasonable distribution of industries in rural areas and their improvement need to be considered.

## 5. Conclusions

On the basis of sample analysis and field investigation of heavy-metal pollution in soil and vegetables, and population and societal survey data in Daye, China, physical and social vulnerability were modeled, and the comprehensive environmental health vulnerability of the area was analyzed with each village as an evaluation unit. The significant result of comprehensive environmental health vulnerability assessment on heavy-metal exposure showed that study areas could be divided into four categories according to the evaluation of environmental health vulnerability. Physical vulnerability indicates the extent of exposure to heavy metals, which is mainly impacted by pollution degree of environmental media and ways for residents to contact pollutants according to the method for calculating physical vulnerability. Areas with high physical vulnerability may be more polluted or more frequently exposed to pollutants by local residents. Therefore, pollution control strategies should be adopted, and education for personal protection awareness and health monitoring should be strengthened. Regions with low physical vulnerability should pay attention to environmental planning and personal protection guidelines. Social vulnerability represents the ability and sensitivity of communities to respond to pollution threats.

The priority control of contaminated areas and pollutants should be effectively identified. According to the analysis of physical vulnerability, the physical vulnerability of villages near mining activities is significantly higher than that of areas far away from mining-affected areas. The physical vulnerability of 10 villages near mining activities and some villages far away from mining activities, such as YQ and SZ, was higher than the acceptable level, which should be listed as the focus of environmental health risk management. In addition, arsenic, cadmium, and copper are the main pollutants in the area, which greatly contribute to physical vulnerability.

In order to promote environmental risk management, it is necessary to give priority to effectively and accurately identifying and controlling pollution areas. Moreover, accuracy means identifying vulnerable populations on the village scale. Furthermore, it is necessary to appropriately adjust the industrial structure and cooperate more to form a complete economic zone.

## Figures and Tables

**Figure 1 ijerph-18-13379-f001:**
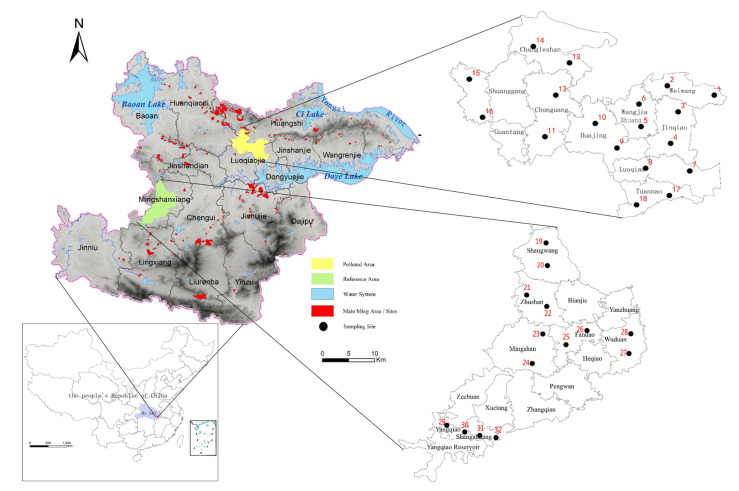
Sampling-point distribution.

**Figure 2 ijerph-18-13379-f002:**
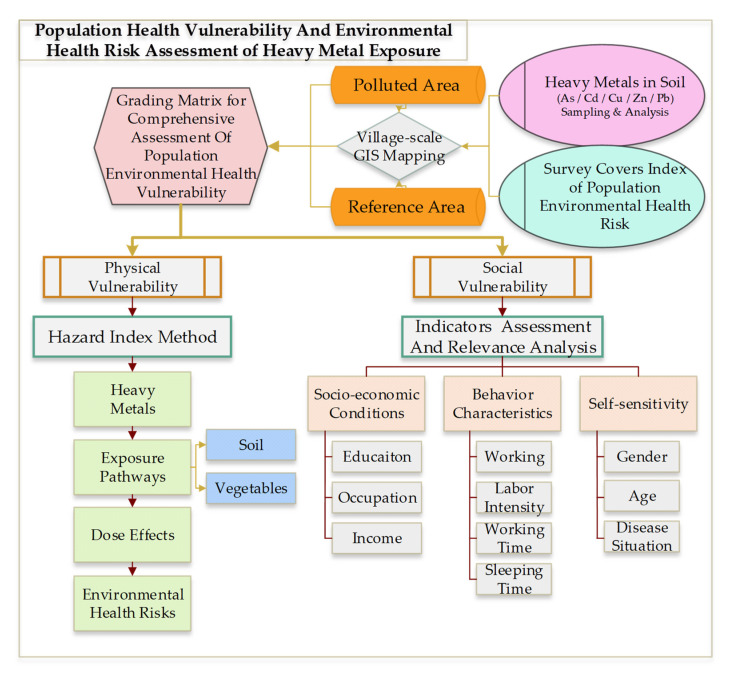
Systematic research solutions for physical–social health vulnerability assessment on heavy metal exposure.

**Figure 3 ijerph-18-13379-f003:**
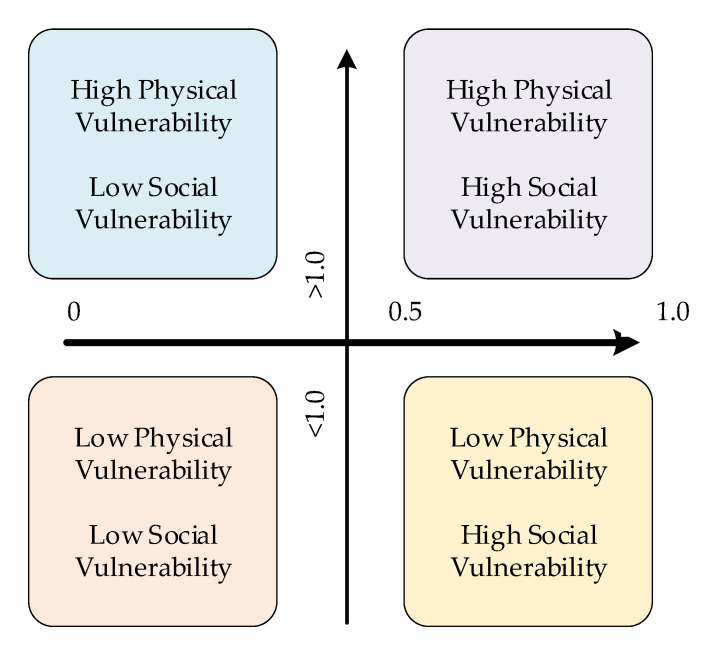
Four-quadrant conceptual frame for environmental health vulnerability assessment.

**Figure 4 ijerph-18-13379-f004:**
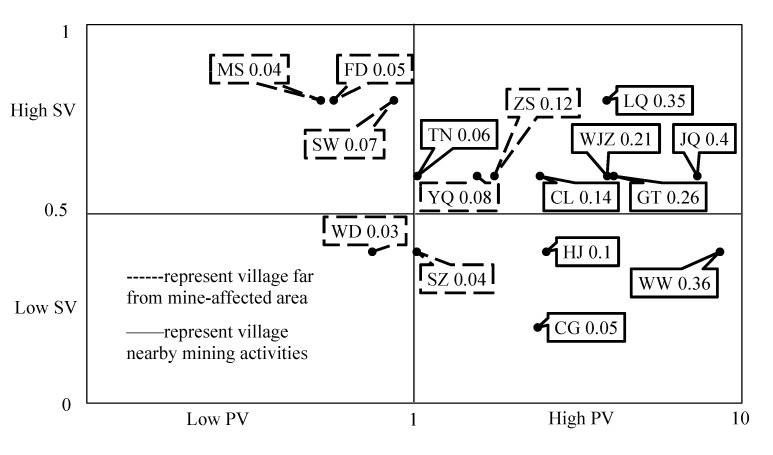
Four-quadrant grading for comprehensive assessment of population health vulnerability (SV = social vulnerability; PV = physical vulnerability).

**Table 1 ijerph-18-13379-t001:** Parameters and their values used to evaluate physical vulnerability.

Parameter	Symbol	Units	Distribution
Vegetable intake rate	IR_v_	mg/day	153.84 ^1^
Soil ingestion rate	IR_o_	mg/day	100
Soil inhalation rate	IR_b_	m^3^/day	20
Exposure frequency	EF_v_ and EF	day/year	365 and 350
Exposure duration	ED	year	24
Body weight	BW	kg	58.53 ^1^
Averaged contact time	AT	day	ED × 365
Particle emission factor	PEF	m^3^/kg	1.36 × 10^9^
Dermal absorption factor	ABS	unitless	0.001
Skin surface area	SA	cm^2^	5218.3 ^2^
Adherence factor	AF	mg/m^2^·day	0.07

^1^ data are actual local population information obtained from survey data; ^2^ data were calculated from the survey.

**Table 2 ijerph-18-13379-t002:** Explanation and quantification of social vulnerability indicators.

Aspects	Indicators	Indicators Explanation	Quantization Method	Weight
Socioeconomic conditions (SEC) 0.343	Education	Divided into 6 categories: undergraduate or above, junior college, secondary school or high school, junior high school, primary school and below, and others	Ratio of qualifications below senior high school	0.109
	Occupation Structure	Divided into 7 categories: agriculture, industry and mining, construction, housewives, self-employed, students, and others	Ratio of occupations with more exposure to heavy-metal pollution	0.117
	Income	Per capita disposable income	Ratio of households below average income	0.117
Receptor characteristics (EB) 0.357	Working conditions	Divided into three categories: good, medium, and poor	Ratio of people in relatively poorer working conditions	0.128
	Labor intensity	Divided into three categories: high, medium, and low	Ratio of people with relatively higher labor intensity	0.086
	Working time	-	Ratio of people working more than 8 h	0.078
	Sleeping time	-	Ratio of people suffering from deficient sleeping time	0.065
Self-sensitivity (SS) 0.300	Gender	Males and females	Female ratio	0.086
	Age	-	Percentage of people younger than 14 or older than 65	0.105
	Disease Situation	Divided into two categories: people who have suffered from disease and those who have not	Percentage of people who have suffered from chronic or major diseases	0.109

Note: the higher the quantified value of the 10 indicators is, the higher the social vulnerability is.

**Table 3 ijerph-18-13379-t003:** Physical vulnerability assessment results of different villages.

Village	HI	Pollution Hazard Index for Different Exposure Pathways				Pollution Hazard Index for Different Heavy Metals				
		HQ_o_	HQ_d_ (10^−4^)	HQ_i_ (10^−6^)	HQ_v_	HI_Cu_	HI_Zn_	HI_As_	HI_Cd_	HI_Pb_
SW	0.86	0.05	4.48	1.28	0.81	0.19	0.07	0.42	0.09	0.09
ZS	1.75	0.04	3.69	1.04	1.71	0.17	0.07	1.27	0.14	0.11
MS	0.52	0.07	4.66	1.72	0.45	0.16	0.05	0.16	0.05	0.10
FD	0.57	0.06	4.42	1.45	0.51	0.12	0.09	0.13	0.16	0.07
WD	0.74	0.06	4.89	1.67	0.68	0.20	0.08	0.34	0.09	0.04
YQ	1.55	0.04	3.29	8.79	1.51	0.12	0.07	1.07	0.14	0.15
SZ	1.01	0.07	5.32	1.76	0.94	0.16	0.07	0.58	0.11	0.09
GT	4.05	0.29	17.9	7.86	3.76	0.19	0.07	3.54	0.10	0.14
CL	2.41	0.12	15.6	2.65	2.29	0.23	0.08	1.22	0.84	0.03
CG	2.37	0.19	19.5	4.55	2.18	0.26	0.07	1.32	0.55	0.16
HJ	2.52	0.17	17.9	4.34	2.34	0.27	0.09	1.38	0.62	0.16
WJZ	3.87	0.10	8.25	2.41	3.78	0.35	0.09	2.76	0.40	0.27
WW	8.55	0.14	18.2	3.14	8.41	0.36	0.13	6.40	1.32	0.34
JQ	7.30	0.12	12.4	2.88	7.18	0.35	0.17	3.91	2.30	0.58
LQ	3.86	0.09	7.26	2.29	3.77	0.37	0.15	2.46	0.31	0.57
TN	1.02	0.07	5.24	1.92	0.94	0.26	0.05	0.58	0.10	0.03

**Table 4 ijerph-18-13379-t004:** Scores of social vulnerability assessment.

Village	Social Vulnerability													Total SV Score
	Socioeconomic Conditions (SEC)			SEC Composite Score	Behavior Characteristics (BE)				RE Composite Score	Self-Sensitivity (SS)			SS Composite Score	
	Education	Occupation Structure	Income		Working Conditions	Labor Intensity	Working Time	Sleeping Time		Gender	Age	Disease Situation		
SW	0.09	0.08	0.12	0.29	0.13	0.09	0.04	0.03	0.28	0.03	0.04	0.05	0.12	0.69
ZS	0.09	0.09	0.07	0.25	0.11	0.07	0.02	0.03	0.23	0.01	0.05	0.05	0.11	0.59
MS	0.11	0.11	0.09	0.31	0.02	0.06	0.07	0.05	0.20	0.01	0.05	0.07	0.13	0.64
FD	0.05	0.09	0.05	0.19	0.08	0.08	0.03	0.06	0.25	0.07	0.08	0.11	0.25	0.69
WD	0.02	0.00	0.02	0.04	0.06	0.04	0.08	0.05	0.24	0.00	0.00	0.07	0.07	0.35
YQ	0.03	0.08	0.05	0.15	0.07	0.04	0.03	0.05	0.19	0.01	0.03	0.06	0.11	0.45
SZ	0.00	0.10	0.04	0.14	0.00	0.03	0.04	0.03	0.09	0.01	0.03	0.07	0.11	0.34
GT	0.02	0.07	0.12	0.20	0.07	0.06	0.02	0.04	0.19	0.03	0.06	0.06	0.15	0.54
CL	0.11	0.03	0.09	0.23	0.02	0.04	0.06	0.03	0.14	0.00	0.06	0.05	0.11	0.49
CG	0.02	0.00	0.00	0.02	0.00	0.00	0.04	0.03	0.07	0.04	0.02	0.05	0.10	0.19
HJ	0.03	0.03	0.05	0.11	0.01	0.03	0.02	0.02	0.08	0.04	0.03	0.07	0.15	0.33
WJZ	0.06	0.07	0.07	0.20	0.00	0.04	0.03	0.06	0.12	0.05	0.03	0.06	0.13	0.46
WW	0.05	0.03	0.06	0.14	0.02	0.00	0.00	0.00	0.02	0.07	0.06	0.07	0.19	0.36
JQ	0.09	0.01	0.07	0.17	0.07	0.06	0.03	0.05	0.22	0.05	0.03	0.00	0.07	0.46
LQ	0.09	0.12	0.09	0.30	0.05	0.02	0.05	0.07	0.18	0.09	0.11	0.10	0.29	0.77
TN	0.03	0.08	0.09	0.20	0.09	0.02	0.00	0.04	0.15	0.04	0.03	0.07	0.15	0.50

**Table 5 ijerph-18-13379-t005:** Relevance analysis of indicators.

Correlation Index		Education	Occupation Structure	Income	Working Condition	Labor Intensity	Working Time	Sleeping Time	Gender	Age	Disease Situation
Education	Pearson Correlation	1	0.217	0.531 ^1^	0.223	0.414	0.278	0.179	0.055	0.453	−0.215
	Significance (bilateral)		0.419	0.034	0.406	0.111	0.297	0.507	0.840	0.078	0.424
Occupation	Pearson Correlation	0.217	1	0.490	0.250	0.331	−0.050	0.387	0.117	0.557 ^1^	0.504 ^1^
	Significance (bilateral)	0.419		0.054	0.350	0.210	0.855	0.139	0.666	0.025	0.046
Income	Pearson Correlation	0.531 ^1^	0.490	1	0.453	0.474	−0.147	0.093	0.073	0.474	−0.066
	Significance (bilateral)	0.034	0.054		0.078	0.064	0.588	0.732	0.787	0.063	0.808
working	Pearson Correlation	0.223	0.250	0.453	1	0.647 ^2^	−0.182	0.157	−0.030	0.118	−0.088
	Significance (bilateral)	0.406	0.350	0.078		0.007	0.501	0.562	0.913	0.664	0.746
Labor Intensity	Pearson Correlation	0.414	0.331	0.474	0.647 ^2^	1	0.183	0.305	−0.233	0.110	−0.120
	Significance (bilateral)	0.111	0.210	0.064	0.007		0.497	0.251	0.386	0.686	0.658
Working Time	Pearson Correlation	0.278	−0.050	−0.147	−0.182	0.183	1	0.411	−0.427	−0.086	0.057
	Significance (bilateral)	0.297	0.855	0.588	0.501	0.497		0.113	0.099	0.750	0.835
Sleeping Time	Pearson Correlation	0.179	0.387	0.093	0.157	0.305	0.411	1	0.183	0.234	0.239
	Significance (bilateral)	0.507	0.139	0.732	0.562	0.251	0.113		0.498	0.384	0.372
Gender	Pearson Correlation	0.055	0.117	0.073	−0.030	−0.233	−0.427	0.183	1	0.576 ^1^	0.339
	Significance (bilateral)	0.840	0.666	0.787	0.913	0.386	0.099	0.498		0.020	0.200
Age	Pearson Correlation	0.453	0.557 ^1^	0.474	0.118	0.110	−0.086	0.234	0.576 ^1^	1	0.502 ^1^
	Significance (bilateral)	0.078	0.025	0.063	0.664	0.686	0.750	0.384	0.020		0.048
Disease Situation	Pearson Correlation	−0.215	0.504 ^1^	−0.066	−0.088	−0.120	0.057	0.239	0.339	0.502 ^1^	1
	Significance (bilateral)	0.424	0.046	0.808	0.746	0.658	0.835	0.372	0.200	0.048	

Note: ^1^ shows significant correlation at 0.05 level (bilateral). ^2^ Shows significant correlation at 0.01 level (bilateral).

## Data Availability

Data sharing does not apply to this article, as no datasets were generated during the current study.

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
