# Peer review of "Rethinking of Environmental Health Risks: A Systematic Approach of Physical—Social Health Vulnerability Assessment on Heavy-Metal Exposure through Soil and Vegetables"

_ijerph, 2021, doi:10.3390/ijerph182413379_

Round 1

Reviewer 1 Report

Although lacking novelty, this work could have been well designed, conducted and presented to provide some information that is useful for readers. However, major flaws were found in research design, data collection, and results discussion. For example, it is true that “contaminated areas” are close to mining activities, but mines also exist near “reference areas”. For heavy metal exposures, “vegetable intake” was used as the only exposure through food consumption, whereas exposure from consumption of other foods like grain and animal foods are not considered. Also, the assumption that people only consume vegetables produced in their own village is unlikely true. While the vulnerability value varied for different villages, no information was provided on how high or low these values are comparing to other regions. Writing of this manuscript is in low quality and numerous errors were found throughout this paper. In my opinion, this manuscript does not reach the publication standards for IJERPH. Therefore, I would suggest rejection of this manuscript.

Author Response

Cover Letter

Dear Reviewer,

First of all, thank you very much for your valuable comments and suggestions. We cherish this modification opportunity given by you and editors as well.

In terms of content, the authors have added several references and rewrote paragraphs to support background, method and results.

In terms of format standardization, we have checked the manuscript and corrected several typo-mistakes and missing words or sentences.

In terms of grammar, we will contact with editor to refine the manuscript thoroughly after this revision if appropriate.

Now, in this letter, the authors revised the manuscript item by item to the opinions of the reviewer, just as bellows:

Reviewer 1

  1. Although lacking novelty, this work could have been well designed, conducted and presented to provide some information that is useful for readers. However, major flaws were found in research design, data collection, and results discussion. For example, it is true that “contaminated areas” are close to mining activities, but mines also exist near “reference areas”.

[Reply] According to reviewers’ comments, the authors have checked the manuscript from beginning to end several times. And several flaws have been found and corrected. For heavy metal exposure problems, soil-vegetable correlation is considered. Firstly, contamination has spacial correlation with polluted areas. Secondly, vegetable for local residents’ ingestion is direct growing in metal contaminated soil. Thirdly, the research proposed to figure out whether there is cross influence of physical vulnerability and social vulnerability in comparable areas that is not far from mining sites. See the full text of the revised manuscript for details.

  1. For heavy metal exposures, “vegetable intake” was used as the only exposure through food consumption, whereas exposure from consumption of other foods like grain and animal foods are not considered. Also, the assumption that people only consume vegetables produced in their own village is unlikely true.

[Reply] In our door-to-door interview and questionnaire survey, a large proportion of local residents have been growing vegetables for self-ingestion, which attract our research attention. And several case study in recent years have shown the rationality of selected soil-vegetable pathway. For local residents, they prefer buying meat and rice from the market than breed poultry or grow rice in their own land. Two representative references of relevant study are shown as follows:

  • Hu, W. Y., Y. Chen, B. Huang and S. Niedermann (2014). "Health Risk Assessment of Heavy Metals in Soils and Vegetables from a Typical Greenhouse Vegetable Production System in China." HUMAN AND ECOLOGICAL RISK ASSESSMENT 20(5): 1264-1280. DOI: 10.1080/10807039.2013.831267.
  • Liu, X. M., Q. J. Song, Y. Tang, W. L. Li, J. M. Xu, J. J. Wu, F. Wang and P. C. Brookes (2013). "Human health risk assessment of heavy metals in soil-vegetable system: A multi-medium analysis." SCIENCE OF THE TOTAL ENVIRONMENT 463: 530-540. DOI: 10.1016/j.scitotenv.2013.06.064.

  1. While the vulnerability value varied for different villages, no information was provided on how high or low these values are comparing to other regions.

[Reply] This manuscript mainly present a systematic modelling method as well as reveal the value difference between contaminated area and reference area. But, unfortunately, the quantified comparison is difficult to conduct to a deeper level due to the data lack and experiments limitation. We have noticed this problem and will continue to improve under appropriate and sufficient support.

  1. Writing of this manuscript is in low quality and numerous errors were found throughout this paper.

[Reply] We are sorry to present the manuscript in an awful way. In the revised version, paragraphs, figures, and charts have been revised carefully as possible as we can.

Thank you again for your help and opinions of academic value.

Best wishes,

Jingcheng Zhou

Research Center for Environment and health, Zhongnan University of Economics and Law

Reviewer 2 Report

The present article, do not present novel research, or novel information. The article presents a few methodologies and models to examinate the risk and influence of heavy metals in human, but the results are simple and within common sense.

The idea and the planning are good, but low details of results are presented.

The grammar and the article structure need to be checked, and the authors must write the article in an easy-to-read way.

There are paragraphs that are repeated several times in the text, this also need to be checked.

Some lines that need to be checked:

- line 44: change “. And researchers…” by “, and researchers…”

- line 46: include commas

- line 90: need comma

- Improve the visualization of Figure 1

- line 130: “vulnerability is defined as….”

- line 136: called

- Figure 2 do not appear in the text

- line 141: add “species” in the phrase “heavy metals species”

- line 145: add comma, “mode, provided….”

- Table 1 do not appear in the text

- line 185: please include verb in the sentence

- line 187 to 190: rewrite the sentence

- Figure 3 do nor appear in the text

- line 232 to 237: rewrite the sentence

- line 242 to 244: rewrite the sentence

- line 247: change “the most”

- line 267: change “the higher the vulnera…”

- line 339: rewrite

- line 368 to 369: rewrite

In my opinion the article not is a novel article, and the grammar and ideas, need to be improved before being accepted by this journal.

Author Response

Cover Letter

Dear Reviewer,

First of all, thank you very much for your valuable comments and suggestions. We cherish this modification opportunity given by you and editors as well.

In terms of content, the authors have added several references and rewrote paragraphs to support background, method and results.

In terms of format standardization, we have checked the manuscript and corrected several typo-mistakes and missing words or sentences.

In terms of grammar, we will contact with editor to refine the manuscript thoroughly after this revision if appropriate.

Now, in this letter, the authors revised the manuscript item by item to the opinions of the reviewer, just as bellows:

Reviewer 2

  1. The present article, do not present novel research, or novel information. The article presents a few methodologies and models to examinate the risk and influence of heavy metals in human, but the results are simple and within common sense.

[Reply] Base on Figure 4, from the aspect of the authors, finding of integrated social-physical vulnerability is presented, villages in contaminated area near mining sites are physical vulnerability predominated, and villages in reference area far from mining sites are social vulnerability predominated although these villages also affected by several mining sites.

  1. The idea and the planning are good, but low details of results are presented.

[Reply] This paper mainly designated a quantitative vulnerability assessment model to evaluate environmental health risk through the heavy metal pathway of soil-vegetable system. Experiments details have been presented in our prior study: 36.   Jun, Y.; Silu, M.; Jingcheng, Z.; Yongwei, S.; Fei, L. Heavy Metal Contamination in Soils and Vegetables and Health Risk As-sessment of Inhabitants In Daye, China. J. Int. Med. Res. 2018, 46, 3374-3387. https://doi.org/10.1177/0300060518758585. Unfortunately, the quantified comparison is difficult to conduct to a deeper level due to the data lack and experiments limitation. We have noticed this problem and will continue to improve under appropriate and sufficient support.

  1. The grammar and the article structure need to be checked, and the authors must write the article in an easy-to-read way.

[Reply] This is a very kind and help notice. The authors will do their best to refine the manuscript smoothly and precisely. As non-native speaker of English, we will prefer to the help of the journal polishing work.

  1. There are paragraphs that are repeated several times in the text, this also need to be checked.

[Reply] Checked.

  1. Some lines that need to be checked:

- line 44: change “. And researchers…” by “, and researchers…”

[Reply] Corrected. In line 44-46, It is expressed as “... to communities [7-9], and researchers in the field of Environmental Justice (EJ) draw into the concept of environmental health vulnerability [10] ...”

- line 46: include commas

[Reply] Corrected.

- line 90: need comma

[Reply] Corrected.

- Improve the visualization of Figure 1

[Reply] High-resolution image of Figure 1, replaces the original image, with 13cm×20.5cm (500 ppi).

- line 130: “vulnerability is defined as….”

[Reply] Corrected. It is expressed as “Vulnerability is defined as ‘the interface between exposure to the physical threats to human and the capacity of people and communities to cope with those threats’ [43].”

- line 136: called.

[Reply] Accepted. In line 145, added “called”.

- Figure 2 do not appear in the text.

[Reply] Corrected. In line 149-151, it is expressed as “Hence, an analytical framework considering the population environmental health vulnerability assessment on heavy metal exposure, is designed as shown in Figure 2.”

- line 141: add “species” in the phrase “heavy metals species”.

[Reply] Corrected. In line 161-164, it is expressed as “The sensitivity of human body to different heavy metal species is different, the exposures of various heavy metals cannot be simply added up, so it is not appropriate to measure physical vulnerability by simply use the heavy metals concentration in the environment media or the average daily doses.”

- line 145: add comma, “mode, provided….”

[Reply] Corrected. In line 165-166, it is expressed as “In this paper, pollution hazard index method in health risk assessment model, provided by USEPA, ...”

- Table 1 do not appear in the text.

[Reply] Corrected. In line 189-191, it is expressed as “Parameters and their values used to evaluate physical vulnerability in equations above are summarized in Table 1.”

- line 185: please include verb in the sentence.

[Reply] Corrected. In line 205, it is expressed as “Explanation and quantification of indicators are shown in Table 2.”

- line 187 to 190: rewrite the sentence.

[Reply] Corrected. In line 209-212, it is expressed as “People with higher incomes feel more satisfied [47] and can afford more local daily consumption. Therefore, we measure economic level of the research areas by the pro-portion of the local people who have reached the per capita disposable income in Daye city.”

- Figure 3 do nor appear in the text.

[Reply] Corrected. In line 237-238, it is expressed as “Based on these two dimensions, a four-quadrant chart is presented for confirming the grade of overall vulnerability, as shown in Figure 3.”

- line 232 to 237: rewrite the sentence.

[Reply] Corrected. In line 259-265, it is expressed as “The physical vulnerability of the villages in the polluted area, was significantly (p<0.05) higher than that in the reference area through each exposure route. The calculated av-erage value of population physical vulnerability in the polluted area and the reference area are 3.99 and 1.00, respectively. Values of physical vulnerability of all villages in the polluted area and ZS, YQ and SZ in the reference area are significantly higher than the upper threshold of 1.0, among which the highest is 8.55.”

- line 242 to 244: rewrite the sentence.

[Reply] Corrected. In line 270-273, it is expressed as “The average physical vulnerability value contribution of different heavy metals shows an order of As (63.9%) > Cd (17.0%) > Cu (8.7%) > Zn (6.8%) > Pb (3.2%). In fact, heavy metals with major contribution to the physical vulnerability are not the same in different exposure routes.”

- line 247: change “the most”.

[Reply] Corrected. In line 274-276, it is expressed as “As contributes the priority among other metals to physical vulnerability value through the exposure routes of soil ingestion and inhalation, reaching at 76.9% and 87.8% respectively.”

- line 267: change “the higher the vulnera…”

[Reply] Corrected. In line 295, it is expressed as “The higher the social vulnerability score is, the lesser social resources the village occupies.”

- line 339: rewrite.

[Reply] Corrected. In line 367-368, it is expressed as ”And indeed, social and physical vulnerability have contribution to the population environmental health risk.”

- line 368 to 369: rewrite.

[Reply] Corrected. In line 395-396, it is expressed as “The results indicated that long term mining activities may influence a rather amount of population in generations and in a large range of areas assuredly.”

Thank you again for your help and opinions of academic value.

Best wishes,

Jingcheng Zhou

Research Center for Environment and health, Zhongnan University of Economics and Law

Reviewer 3 Report

The manuscript  titled : “Rethinking of environmental health risks: A systematic approach of physical-social health vulnerability assessment on heavy metal exposure” "assesses, soil and plant pollution, and population and social characteristics data by correlating them with environmental health vulnerability assessments.

In order for the work to be published in my opinion it is necessary a major revision  in order to clarify some points and provide other informations. In addition the work is quite complex and for understanding to a wide readership the authors should be a little clearer. What I find in this work is that the authors take some things for granted but it is a bit difficult to understand the work. Therefore I invite the authors to accompany the reader to the understanding of the work. I want to see a revised version in which I will be clearer about the content and the message that the authors want to give.

Other observations are the following:

  • LINE 44 THE SENTENCE “from both chemical and non-chemical stressors to communities. And Researchers in the…. IT IS UNCLEAR

  • Lines 53……“In order to evaluate the environmental health vulnerability, there are four prominent evaluation index systems that are assisted by GIS mapping tools, which are: the Cumulative Environmental Hazard Inequality Index (CEHII) [17, 18], the Environmental Justice Screening Method (EJSM) [19, 20] and its associated Climate Change Vulnerability Screening  Method (CCVSM) [21], the Cumulative Environmental Vulnerability Assessment (CEVA)  [22] and the California Community Environmental Health Screening Tool (CalEnvi-roScreen) [23]. The indicators of such methods are so broad that easily ignore data on pollution exposure, in addition, these indicators are also often used at the state and provincial levels, so the evaluation units are too large to accurately reflect small-scale diffeences such as villages or particular tantamount areas”.

The authors should be more cautious in these statements

  • The authors should even better explain what is meant by vulnerability and how vulnerability indices are calculated
  • Pollution affects reproductive health. Human semen is now considered an early sentinel of an organism's health status. Human semen is very susceptible to environmental pollutants.

I recommend the authors to read and quote the following work:

Molecular Alterations in Spermatozoa of a Family Case Living in the Land of Fires. A First Look at Possible Transgenerational Effects of Pollutants  DOI: 10.3390/ijms21186710

  • “Psychological and physiological conditions of people after 13 years tend to be mature [44], while the national legal retirement age is 65 years old [45]. We have reason to think that people aged 14-65 have better physical and psychological qualities to resist and defend potential threats. Therefore, the proportion of the population under 14 years old and over 65 years old in this area is regarded as the age with higher  vulnerability”

I'm not clear on this reasoning

  • “Low physical vulnerability is defined as a situation where the pollution hazard index is less than 1, because it is acceptable that the pollution hazard index is lower than 1 according to the relevant research of USEPA. And social vulnerability scores ranged from 0 to 1, so low social vulnerability is defined as a situation where the score of social vulner ability is less than 0.5. The idea of conceptual frame is used to classify vulnerability into  four categories, and different villages will be located in different areas of the quadrant”.

Better explain the reasons for this classification

  • The authors should mention the most relevant effects of arsenic and cadmium. Still referring to reproductive health I suggest the authors to read and quote the following work that deals with the effects of cadmium on proteins that organize the sperm chromatin. Although the work is done on mussels similar effects with various metals are also being seen on humans

Alterations in the properties of sperm protamine-like II protein after exposure of Mytilus galloprovincialis (Lamarck 1819) to sub-toxic doses of cadmium. doi: 10.1016/j.ecoenv.2018.11.069.

  • There is work in the literature on the ability of Feijoa sellowiana extracts to counteract the deleterious effects of mercury on red blood cells. Also always extracts of this plant have shown antibacterial and antioxidant activity. I would like to know if the authors evaluated these activities in the foods considered and if, if so, they found greater or lesser activity in those in polluted areas. I would like the authors to argue on this as well

Author Response

Cover Letter

Dear Reviewer,

First of all, thank you very much for your valuable comments and suggestions. We cherish this modification opportunity given by you and editors as well.

In terms of content, the authors have added several references and rewrote paragraphs to support background, method and results.

In terms of format standardization, we have checked the manuscript and corrected several typo-mistakes and missing words or sentences.

In terms of grammar, we will contact with editor to refine the manuscript thoroughly after this revision if appropriate.

Now, in this letter, the authors revised the manuscript item by item to the opinions of the reviewer, just as bellows:

Reviewer 3

  1. The manuscript titled: “Rethinking of environmental health risks: A systematic approach of physical-social health vulnerability assessment on heavy metal exposure” "assesses, soil and plant pollution, and population and social characteristics data by correlating them with environmental health vulnerability assessments.

[Reply] Corrected. We have modified the tile precisely with “Rethinking of environmental health risks: A systematic approach of physical-social health vulnerability assessment on heavy metal exposure through soil and vegetables”, by hints of two representative references of relevant study are shown as follows:

(1)   Hu, W. Y., Y. Chen, B. Huang and S. Niedermann (2014). "Health Risk Assessment of Heavy Metals in Soils and Vegetables from a Typical Greenhouse Vegetable Production System in China." HUMAN AND ECOLOGICAL RISK ASSESSMENT 20(5): 1264-1280. DOI: 10.1080/10807039.2013.831267.

(2)   Liu, X. M., Q. J. Song, Y. Tang, W. L. Li, J. M. Xu, J. J. Wu, F. Wang and P. C. Brookes (2013). "Human health risk assessment of heavy metals in soil-vegetable system: A multi-medium analysis." SCIENCE OF THE TOTAL ENVIRONMENT 463: 530-540. DOI: 10.1016/j.scitotenv.2013.06.064.

  1. In order for the work to be published in my opinion it is necessary a major revision in order to clarify some points and provide other information. In addition, the work is quite complex and for understanding to a wide readership the authors should be a little clearer. What I find in this work is that the authors take some things for granted but it is a bit difficult to understand the work. Therefore, I invite the authors to accompany the reader to the understanding of the work. I want to see a revised version in which I will be clearer about the content and the message that the authors want to give.

[Reply] Accepted. We have revised languages, figures, tables and paragraphs to meet the acquirement of the reviewer as possible as we could. Recommended references from the reviewer and new supportive references by literature retrieval are added in revised manuscript.

  1. LINE 44 THE SENTENCE “from both chemical and non-chemical stressors to communities. And Researchers in the…. IT IS UNCLEAR

[Reply] Corrected. In line 44-45, it is expressed as “... both chemical mixtures and combination of chemical and non-chemical stressors to communities [7-9] ...”. The expression of “chemical mixtures”, and “combination of chemical and non-chemical stressors”, are referenced according to the literature [7-9].

  1. Lines 53……“In order to evaluate the environmental health vulnerability, there are four prominent evaluation index systems that are assisted by GIS mapping tools, which are: the Cumulative Environmental Hazard Inequality Index (CEHII) [17, 18], the Environmental Justice Screening Method (EJSM) [19, 20] and its associated Climate Change Vulnerability Screening Method (CCVSM) [21], the Cumulative Environmental Vulnerability Assessment (CEVA)  [22] and the California Community Environmental Health Screening Tool (CalEnviroScreen) [23]. The indicators of such methods are so broad that easily ignore data on pollution exposure, in addition, these indicators are also often used at the state and provincial levels, so the evaluation units are too large to accurately reflect small-scale differences such as villages or particular tantamount areas”. The authors should be more cautious in these statements.

[Reply] Corrected. In line 53-63, it is expressed as “In order to evaluate the environmental health vulnerability, there are four prominent evaluation index systems that are assisted by GIS mapping tools, which include the Cumulative Environmental Hazard Inequality Index (CEHII) [17, 18], the Environ-mental Justice Screening Method (EJSM) [19, 20] with its associated Climate Change Vulnerability Screening Method (CCVSM) [21], the Cumulative Environmental Vul-nerability Assessment (CEVA) [22], and the California Community Environmental Health Screening Tool (CalEnviroScreen) [23]. The indicators of such methods are of-ten used at the state or provincial levels, thus the evaluation units are medium-level oriented, which hint us to prefer a much finer level, such as villages or particular tan-tamount areas, to accurately reflect small-scale differences.”

  1. The authors should even better explain what is meant by vulnerability and how vulnerability indices are calculated.

[Reply] Accepted. In line 139-156, two paragraph are explained. “Vulnerability is defined as ‘the interface between exposure to the physical threats to human and the capacity of people and communities to cope with those threats’ [43]. For environmental risk management, the assessment of environmental health vulnerability needs to consider two aspects: one is the heavy metal exposure and possible health of communities or groups, called physical vulnerability [44], and another is the comprehensive measurement of the sensitivity, coping ability, adaptability and resilience of communities in the face of threats, namely called social vulnerability [45].

We suppose that integrated population environmental health vulnerability can be divided into these two aspects, which are physical vulnerability and social vulnerability, then research for a certain population in case study to figure out how to evaluate the vulnerability mentioned above. Hence, an analytical framework considering the population environmental health vulnerability assessment on heavy metal exposure, is designed as shown in Figure 2. It is worth mentioning that oral intake and soil expo-sure are two specific pathways to conduct this research. Moreover, by a preliminary investigation, we found out that a majority of local residents feed vegetables growing in local soil, which may have been affected by heavy metal exposure, and mainly grow them for self-consumption. Therefore, what is more important is that contaminated soil and vegetable intake has correlation.”

  1. Pollution affects reproductive health. Human semen is now considered an early sentinel of an organism's health status. Human semen is very susceptible to environmental pollutants. I recommend the authors to read and quote the following work: Molecular Alterations in Spermatozoa of a Family Case Living in the Land of Fires. A First Look at Possible Transgenerational Effects of Pollutants. DOI: 10.3390/ijms21186710.

[Reply] Accepted. In line 63-67, it is added as “Pollution affects human health, what is more, pollution affects reproductive health [24, 25]. Heavy metals, being considered as critical kind of pollutants, i.e. cadmium and arsenic, of exposition sources, resorption pathways and organ damage processes have been focused for a long time [26-29].”

  1. “Psychological and physiological conditions of people after 13 years tend to be mature [44], while the national legal retirement age is 65 years old [45]. We have reason to think that people aged 14-65 have better physical and psychological qualities to resist and defend potential threats. Therefore, the proportion of the population under 14 years old and over 65 years old in this area is regarded as the age with higher vulnerability”. I'm not clear on this reasoning.

[Reply] Accepted. Several sentences in line 214-222 are modified to explain the reason: “Psychological and physiological conditions of a person after 13 years tend to be mature [50], while Chinese legal retirement age is 65 years old in common [51]. We suppose to think that people aged 14-65 are well-balanced and usually be able to resist potential social or physical threats. On the contrary, the people under 14 years old or over 65 years old in this area, seems too young or too old to face threats off balance according to their capabilities. Therefore, people under 14 years old or over 65 years old, which is regarded as the age with much higher vulnerability. Besides, it is easy to identify the vulnerable population in various indicators according to previous literature research, such as education level, occupation [52], working environment, labor intensity, gender [53] and disease.”

  1. “Low physical vulnerability is defined as a situation where the pollution hazard index is less than 1, because it is acceptable that the pollution hazard index is lower than 1 according to the relevant research of USEPA. And social vulnerability scores ranged from 0 to 1, so low social vulnerability is defined as a situation where the score of social vulnerability is less than 0.5. The idea of conceptual frame is used to classify vulnerability into four categories, and different villages will be located in different areas of the quadrant”. Better explain the reasons for this classification.

[Reply] Accepted. In line 239-247, we revised the sentence as: “Physical vulnerability is defined as a threshold when the pollution hazard index is less than 1.0, else more than 1.0. According to the relevant research of USEPA, it is reason-able to believe that the pollution hazard index is lower than 1. In addition, for social vulnerability, in a common sense, appropriate discontinuous scores of top to bottom limitation is 0 to 1.0. However, integrate discrete-value with continuous-value may cause conflict in general vulnerability assessment, so social vulnerability is also de-fined as a threshold when the score of social vulnerability is less than 0.5 to 0, else more than 0.5 to 1.0. Therefore, a quantified conceptual frame can be used to classify vulnerability into four categories, and different villages will be sorted in different areas of the quadrant.”

  1. The authors should mention the most relevant effects of arsenic and cadmium. Still referring to reproductive health I suggest the authors to read and quote the following work that deals with the effects of cadmium on proteins that organize the sperm chromatin. Although the work is done on mussels similar effects with various metals are also being seen on humans: Alterations in the properties of sperm protamine-like II protein after exposure of Mytilus galloprovincialis (Lamarck 1819) to sub-toxic doses of cadmium. doi: 10.1016/j.ecoenv.2018.11.069.

[Reply] Accepted. Please see the reply of 6 above.

  1. There is work in the literature on the ability of Feijoa sellowiana extracts to counteract the deleterious effects of mercury on red blood cells. Also always extracts of this plant have shown antibacterial and antioxidant activity. I would like to know if the authors evaluated these activities in the foods considered and if, if so, they found greater or lesser activity in those in polluted areas. I would like the authors to argue on this as well.

[Reply] For this question, we are not able to find the counteract effects of Feijoa sellowiana or similar extracts in foods due to the lack of molecule research background. And in the villages of polluted area, seven kind of vegetables (see line 120-125, “cowpea (Vigna unguiculata (Linn.) Walp), water spinach (Ipomoea aquatica Forsk), ama-ranth (Amaranthus tricolor L.), sweet potato leaves (Ipomoea batatas Lam), tomato (Lycopersicon esculentum Miller), eggplant (Solanum melongena Linn), and pepper (Capsicum annuum Linn. var. gros-sum (L.) Sendt). The pretreatment and analysis of soil samples and vegetable samples were based on our previous research data, which has been analyzed in detail in Jun et al [42]”.) growing in local field, which are also the main vegetables for human ingestion by local residents in our sampling and survey period.

Thank you again for your help and opinions of academic value.

Best wishes,

Jingcheng Zhou

Research Center for Environment and health, Zhongnan University of Economics and Law

Round 2

Reviewer 1 Report

Thanks for the revision/clarification, the revised manuscript is a better version and addressed most of the comments by reviewers. I would recommend publication after minor revision as detailed below. 

1) In the manuscript please include the explanation listed in the response letter about "residents buy rice and meat from market and consume vegetables they grow". 

2) Further edits/corrections/polish are required to reach publication standards in the journal of IJERPH. For example, the sentence at Line 160 should be rewritten; Line 161, "kind of" should be deleted, better edit the sentence to "have attracted scientific attention for decades"; Line 322, delete "we suppose that"; Line 436, delete "we suppose to think that"; Line 518, delete "this". Also, please pay attention to tense used in the writing.

Author Response

Cover letter

Dear Reviewer,

Thank you for your conscientious help. The authors have checked the manuscript again to fulfill your professional advices.

Now, in this letter, the authors revised the manuscript item by item to the comments of the reviewer, just as bellows:

Reviewer’ Comments

  • In the manuscript please include the explanation listed in the response letter about "residents buy rice and meat from market and consume vegetables they grow".

[Authors’ Reply] Accepted. This explanation is added in “2.2.1 Heavy Metal Pollution Data”, rice mainly refers to paddy rice, with supplement of wheat flour, and meat mainly include: pork, chicken (as food), mutton, beefs. These types have been added in line 123-126 of the manuscript as well.

After second revision by authors, the revised manuscript has passed English Editing by professionals. And in this latest revision, the content is revised as “A large number of local residents buy rice and meat from markets, and consume vegetables that they grow in their own field. Moreover, rice mainly refers to paddy rice, with supplements of wheat flour, and meat mainly includes pork, chicken (as food), mutton, and beef.” in “2.2.1. Heavy-Metal Pollution Data”.

  • Further edits/corrections/polish are required to reach publication standards in the journal of IJERPH. For example, the sentence at Line 160 should be rewritten; Line 161, "kind of" should be deleted, better edit the sentence to "have attracted scientific attention for decades"; Line 322, delete "we suppose that"; Line 436, delete "we suppose to think that"; Line 518, delete "this". Also, please pay attention to tense used in the writing.

[Authors’ Reply] Accepted. In the second revision, the authors have checked the whole content of the paper. Please see the revised version of the paper for details. Furthermore, our manuscript will undergo extensive English revision with the help of the editing service.

The paper also has been go through the plagiarism check and professional English Editing Service. Please see the attachment.

We deeply thank you for your time and all the helping effort!

Best regards,

Jingcheng Zhou

Zhongnan University of Economics and Law

Reviewer 3 Report

The authors responded to all of my requests. I accept the work in the present form

Author Response

Cover letter

Dear Reviewer,

Thank you for your conscientious help and professional advices. The authors have checked the manuscript over again to improve the paper to the standard of the scientific journal.

Now, in this letter, the authors revised the manuscript item by item to the comments of the reviewer, just as bellows:

Reviewer’ Comments

The authors responded to all of my requests. I accept the work in the present form.

Authors’ Reply

Thank you very much. In the second revision, the authors have checked the whole content of the paper. Please see the revised version of the paper for details. In the second revision, the authors have checked the whole content of the paper. Please see the revised version of the paper for details.

After second revision, the latest revised manuscript also has been go through the plagiarism check and professional English Editing Service. Please see the attachment.

We deeply thank you for your time and all the helping effort!

Best regards,

Jingcheng Zhou

Zhongnan University of Economics and Law
